# Transparent Glasses and Glass-Ceramics in the Ternary System TeO_2_-Nb_2_O_5_-PbF_2_

**DOI:** 10.3390/ma14020317

**Published:** 2021-01-09

**Authors:** Juliana Santos Barbosa, Gislene Batista, Sylvain Danto, Evelyne Fargin, Thierry Cardinal, Gael Poirier, Fabia Castro Cassanjes

**Affiliations:** 1Institute of Science and Technology, Federal University of Alfenas, Campus Poços de Caldas, MG CEP, Poços de Caldas 37715-400, Brazil; julianasantos92@hotmail.com.br (J.S.B.); gislene.batista@sou.unifal-mg.edu.br (G.B.); fabia.cassanjes@unifal-mg.edu.br (F.C.C.); 2Institut de Chimie de la Matière Condensée de Bordeaux—ICMCB, Université de Bordeaux, 87 Avenue du Dr. Schweitzer, F-33608 Pessac, France; sylvain.danto@u-bordeaux.fr (S.D.); evelyne.fargin@u-bordeaux.fr (E.F.); Thierry.Cardinal@icmcb.cnrs.fr (T.C.)

**Keywords:** glass, glass-ceramic, fluoride, tellurite, crystallization, europium

## Abstract

Transparent fluorotellurite glasses were prepared by melt-quenching in the ternary system TeO_2_-Nb_2_O_5_-PbF_2_. The synthesis conditions were adjusted to minimize fluorine loss monitored as HF release. It was found that 10 mol% of Nb_2_O_5_ is the optimum content for PbF_2_ incorporation up to 35 mol% in the tellurite matrix without loss of glass forming ability. Such glass compositions exhibit a wide optical window from 380 nm to about 6 μm. Crystallization properties were carefully investigated by thermal analysis and compositions with higher PbF_2_ contents exhibit preferential precipitation of lead oxyfluoride Pb_2_OF_2_ at lower temperatures. The lead oxyfluoride crystallization mechanism is also governed by a volume nucleation, barely reported in tellurite glasses. Eu^3+^ doping of these glass compositions also promotes a more efficient nucleation step under suitable heat-treatments, resulting in transparent Eu^3+^-doped glass-ceramics whereas undoped glass-ceramics are translucent. Finally, Eu^3+^ spectroscopy pointed out a progressive, more symmetric surrounding around the rare earth ions with increasing PbF_2_ contents as well as higher quantum efficiencies. These new fluorotellurite glass compositions are promising as luminescent hosts working in the middle infrared.

## 1. Introduction

Oxyfluoride glasses are well-known in glass science for their famous combined properties, taking advantage of both oxide and fluoride compounds as in composite materials [1,2,3,4,5]. The final goal of oxyfluoride glasses is usually related to optical applications since fluoride compounds bring higher optical performances in key areas such as luminescence quantum efficiencies or very low optical attenuations [6], whereas oxide glassy materials are cheaper, easier to prepare, and more thermally and chemically stable for large pieces production or fiber drawing design [7,8,9,10,11,12,13]. Such optical properties can be further improved when heavy metal fluoride nanocrystals are grown in the oxyfluoride glass by suitable heat-treatments. In fact, the fluoride crystalline material governs the optical response whereas the oxide glassy matrix ensures the mechanical and thermal properties of the whole material [14,15,16,17,18,19]. Spatially-controlled precipitation of crystalline fluorides can also induce a local refractive index variation in the so-called photothermorefractive glasses with important associated applications such as diffraction gratings or holographic recording [20,21,22,23,24].

However, some important intrinsic optical characteristics of oxyfluoride glasses and glass-ceramics are also limited by the oxide glass former and cannot be overcome by simply adding fluoride compounds. In particular, extended optical transparency in the middle infrared, which is a key parameter for optical devices working above 3 μm, is related to harmonics of the fundamental molecular motions and is usually governed by the high frequency modes of the oxide molecular units [25]. In this case, well-known fluorosilicate, fluoroborate, and fluorophosphate glasses and glass-ceramics do not meet these technical requirements and other oxide formers with lower fundamental frequency motions, i.e., with heavier cations, must be considered. Among the known oxide glass formers, tellurium oxide TeO_2_ is the best candidate for such purposes with the wider optical window up to 6 μm [26]. Moreover, the higher atomic number of tellurium promotes other interesting optical characteristics in tellurite glasses such as lower phonon energies and higher polarizabilities, explaining why these glasses display strong interest as rare earth hosts [26,27,28,29,30,31,32] and third order non-linear optical materials [26,33,34,35,36,37].

However, despite the promising potentialities of transparent fluorotellurite glasses and glass-ceramics, such materials were less investigated when compared with fluorosilicates, fluorophosphates, and fluorogermanates since large incorporations of heavy metal fluoride compounds usually induce glass devitrification. Another critical drawback of tellurite glasses for transparent glass-ceramics production is related to their dominant surface-induced crystallization. Most reported works on this subject are devoted to high alkaline [38,39] and zinc fluoride [40,41,42,43,44,45] contents in transparent tellurite glasses without a loss of the glass forming ability. Only a few works reported on transparent tellurite glasses with high heavy metal fluoride contents such as BaF_2_ and PbF_2_ up to 30 mol% [46,47,48,49,50]. A single report describes tellurite glasses with up to 50 mol% of PbF_2_, even if the glassy state of the resulting samples obtained by melt-quenching is not demonstrated [51]. Fluorotellurite glass-ceramics containing calcium and barium fluoride crystalline nanocrystals were also reported but with a final translucent aspect associated to crystallites scattering [52,53,54,55].

Based on these considerations, this work intended to investigate glass formation as well as structural, optical, and crystallization properties in the ternary system TeO_2_-Nb_2_O_5_-PbF_2_ where niobium oxide acts as an intermediate compound used to increase the glass thermal stability for lead fluoride addition. Loss of starting compounds by evaporation or decomposition were investigated and synthesis parameters optimized. The effect of europium doping on the crystallization mechanisms of fluorine-based crystalline phases was investigated for functional transparent oxyfluoride glass-ceramics. Eu^3+^ spectroscopy was also used as a structural probe for a better understanding of the structural changes versus composition and crystallization state.

## 2. Materials and Methods

Glass formation was investigated in the ternary system TeO_2_-Nb_2_O_5_-PbF_2_. For this preliminary investigation, starting compounds TeO_2_ (Sigma-Aldrich, Darmstadt, Germany, 99+%), Nb_2_O_5_ (Sigma-Aldrich, Darmstadt, Germany 99.9%) and PbF_2_ (Sigma-Aldrich, Darmstadt, Germany, 99.9%) were ground in an agate mortar, melted in a gold crucible at 890 °C for 10 min and quenched in a cold stainless-steel mold. In this first step, the identification of the final material as glass, glass-ceramic, or ceramic was determined by the visual aspect and transparency of the sample as depicted in Figure 1. Glass compositions identified in our ternary system and labeled as xNbyPb where x and y represent the Nb_2_O_5_ and PbF_2_ contents (mol%) respectively along with their characteristic temperatures (Tg: glass transition temperature, Tx: temperature at the onset of crystallization, Tc: temperature at the peak of crystallization) are resumed in Table 1. Glasses in the series (100-y)TeO_2_-10Nb_2_O_5_-yPbF_2_ (0 ≤ y ≤ 30) were selected for further investigations. For preparing larger glass pieces with good optical quality and mechanical properties, a detailed study of mass loss of the starting compounds during heating was performed to define suitable synthesis parameters (melting temperature and time): a starting powder mixture of composition 65TeO_2_-10Nb_2_O_5_-25PbF_2_ was analyzed using a FTIR-coupled thermogravimetric balance. These TG-FTIR measurements were performed with a TG Setsys “Evo” apparatus (Setaram Instrumentation, Caluire-et-Cuire, France) coupled on-line to a FTIR Thermo Nicolet 380 spectrometer equipped with corrosive gas version (ThermoFisherScientific, Waltham, MA, USA). In all experiments, the carrier gas was Nitrogen (Air Liquide—Alphagaz 2, with a purity of: O_2_ < 100 ppb, H_2_O < 500 ppb and total hydrocarbon (CxHy) < 100 ppb) at a flow of 50 mL/min^−1^.

Heating was carried out at a rate of 5 K·min^−1^, and a platinum crucible was used as the sample holder. TG-FTIR results were obtained using a stainless steel cell equipped with a BaF_2_ optical window which allows infrared detection in the range of 4400–740 cm^−1^. Evolved gaseous species were passed through a temperature gas transfer line equipped with a PFA (PerFluoroAlkoxy) capillary. Both gas sample cell and gas transfer line were maintained at a constant temperature of 443 K to avoid cold surfaces and thus to prevent the condensation of evolved gases. FTIR spectra were collected with a resolution of 2 cm^−1^, 32 scans being collected per spectrum. Based on these thermogravimetric results and related released species (described in the results and discussion section), melting temperatures and times were optimized to the lower values resulting in a homogeneous melt before casting. According to these considerations, starting powders were ground in an agate mortar for 15 min and melted in a covered gold crucible for specific melting times and temperatures as follows: 890 °C during 10 min for 10Nb0Pb, 890 °C during 5 min for 10Nb10Pb, 870 °C during 5 min for 10Nb20Pb and 850 °C during 5 min for 10Nb30Pb. Then, the homogeneous melt was quenched in a stainless-steel mold preheated at 40 °C below the glass transition temperature and annealed at this temperature for 4 h before slow cooling to room temperature. Eu^3+^-doped compositions were obtained following the same methodology (Eu_2_O_3_ from Sigma-Aldrich, São Paulo, Brazil, 99.99%) using low doping contents of 0.1% in order to avoid energy transfer processes between Eu^3+^ ions. The glasses were cut and polished in both sides using polishing sandpaper until a 1mm-thick glass sample is obtained. Glass-ceramics of selected compositions were prepared by heat-treatment of the pristine glasses at the onset of crystallization (Tx) for 1 h and labeled as xNbyPb-TT for undoped samples and xNbyPb-Eu-TT for Eu^3+^-doped samples. DSC (Differential Scanning Calorimetry) measurements were performed on bulk samples using a DSC Maia F200 calorimeter (Netzsch-Gerätebau GmbH, Selb, Germany) between 200 °C and 450 °C, since higher temperatures promote mass losses, using a heating rate of 10 °C/min in sealed aluminum pans under N_2_ flowing atmosphere. The crystallization study was performed by DSC analysis on powder samples with controlled grain sizes. The glass compositions were ground in an agate mortar and the final powder separated using nylon sieves with mesh 63 μm, 53 μm, 45 μm, 38 μm, and 20 μm. Powders with grain sizes above 63 μm and below 20 μm were not analyzed. Raman spectra were collected in backscattering mode on polished undoped glasses between 200 cm^−1^ and 1000 cm^−1^ with a LabRam Micro-Raman (Horiba Jobin-Yvon, Kyoto, Japan) working with a He-Ne laser at 632.8 nm.

X-ray diffraction measurements were performed on powder samples using a Rigaku ultima IV diffractometer (Rigaku Corporation, Tokyo, Japan) working at 40 KV and 30 mA between 10° and 80° in continuous step mode of 0.02°/s. The crystalline phases were identified according to X-ray powder diffraction patterns (PDF file). UV-visible transmission and attenuation spectra were recorded between 350 nm and 2500 nm at 0.5 nm/s using a Cary 7000 spectrophotometer (Agilent Technologies, Santa Clara, CA, USA). Transmission infrared spectra were collected between 650 cm^−1^ and 4000 cm^−1^ using a spectrometer Cary 630 (Agilent technologies). Finally, the emission spectra of Eu^3+^-doped samples were recorded in a spectrofluorimeter model Fluorolog FL3-221 (Horiba Jobin Yvon, Kyoto, Japan), equipped with CW xenon flash lamp and a photodiode detector (HORIBA PPD-850). The excited state lifetime values of Eu^3+^ (^5^D_0_ state) were determined by exponentially fitting the PL decay curves obtained from excitation with a pulsed flash lamp.

## 3. Results and Discussion

The glass forming ability in the ternary system TeO_2_-Nb_2_O_5_-PbF_2_ is depicted in Figure 1. As expected, in a preliminary analysis of the glass-forming domain, it appears that Nb_2_O_5_ behaves as an intermediary compound since it stabilizes higher lead fluoride contents when compared to the binary system TeO_2_-PbF_2_. In fact, 10 and 15 mol% of Nb_2_O_5_ allows incorporating 35 and 30% of PbF_2_ respectively. On the other hand, high niobium oxide contents increase the melting point and viscosity of the final batch and should require higher melting times and temperatures. For this reason, glass-forming ability was not investigated beyond 15 mol% Nb_2_O_5_ since the starting compounds do not melt properly under our synthesis conditions. In this sense, the series (100 − y)TeO_2_-10Nb_2_O_5_-yPbF_2_ seems to be the most promising for further investigations. One should note that such high PbF_2_ contents in tellurite glasses are barely reported and these reports do not give details of the synthesis conditions, final glassy state and fluoride losses [50,51]. DSC measurements were performed on all glass compositions as presented in Figure 2 with characteristic temperatures resumed in Table 1.

These thermal results highlight the opposite intermediary and modifier behaviors of Nb_2_O_5_ and PbF_2_ on the tellurite glass network since niobium insertion progressively increases the glass transition temperature along the three series whereas higher PbF_2_ contents clearly promote a Tg decrease and lower glass thermal stability against devitrification within each series. Such behavior can be understood by insertion of NbO_6_ units inside the tellurite network built from TeO_4_ and TeO_3_, resulting in cross-linking between the tellurite chains and higher overall connectivity. On the other hand, fluorine addition should promote the formation of terminal Te-F and Nb-F bonds i.e., a less-connected glass structure. Besides these considerations, an intense sharp crystallization event is detected in series 5NbyPb and 10NbyPb at low temperatures for high PbF_2_ contents whereas another weak and broad crystallization peak occurs at higher temperatures. Since such an event is related to the lead fluoride content, one can suggest that this crystalline phase contains fluorine and, in a first attempt, is attributed to a fluoride or oxyfluoride compound. In the series 15NbyPb, no crystallization is detected up to 450 °C, in agreement with the stabilizing behavior of niobium, while investigations at higher temperatures were not performed as the mass loss was detected beyond 450 °C. Since this work is devoted to the study of niobium-tellurite glasses with high PbF_2_ contents as well as the precipitation of fluorine-based crystalline phases for technological glass-ceramics, the series 10NbyPb was selected for further thermal, structural and optical investigations.

Owing to the well-known sublimation and evaporation tendencies of tellurium oxide TeO_2_ and lead fluoride PbF_2_ under heating, this work first focused on the understanding of chemical losses during melting of the starting powders in the ternary system TeO_2_-Nb_2_O_5_-PbF_2_. FTIR coupled-thermogravimetric analyzes were performed on a powder mixture of composition 65TeO_2_-10Nb_2_O_5_-25PbF_2_ in an open crucible from room temperature to 850 °C as a first attempt to understand the origin of mass losses during heating and melting, as depicted in Figure 3a. Upon a heating rate of 10 °C/min from room temperature, one could note that mass loss of the starting powders begins around 400 °C and reaches about 10% at 850 °C with a continuous and linear mass loss at such isotherm.

Coupled FTIR measurements of the evolved gases during thermogravimetry detected increasing amounts of released hydrogen fluoride HF from 400 °C to the end of the measurement, identified by a clear FTIR signature around 4000 cm^−1^ as highlighted in Figure 3b. Another FTIR signal centered at 1028 cm^−1^ appears at the beginning of mass loss between 350 °C and 650 °C but could not be attributed with specific reported molecular vibrations related to our starting compounds. However, a report on the reaction of molten Nb_2_O_5_ with gaseous F_2_ identified several gaseous niobium oxyfluoride of general formula NbOxFy, which are precisely released between 350 °C and 550 °C. In this sense, one can suggest that fluoride ions from lead fluoride react with niobium oxide in this temperature range. Such complex niobium oxyfluoride species could be at the origin of our IR signal at 1028 cm^−1^ [56].

These results highlighted a significative mass loss of fluorine upon heating, justifying our choice to use the lowest melting temperatures and times able to produce a homogeneous melt for each composition as detailed in the experimental section. Keeping in mind that all compositions containing PbF_2_ in the 10NbyPb series were melted for 5 min between 850 °C and 890 °C, thermogravimetric data of the isotherm allowed to estimate a mass loss around 1.2% for a 5 min heating step at 850 °C. Thus, for a starting powder mixture of composition 65TeO_2_-10Nb_2_O_5_-25PbF_2_ and considering that all fluorine is lost as HF produced by reaction between H_2_O or hydroxyls and PbF_2_, a 1.2% mass loss of the starting mixture gives a final molar composition 65TeO_2_-10Nb_2_O_5_-20PbF_2_-5PbO.

Raman spectroscopy performed on the polished undoped glasses (Figure 4) in the series 10NbyPb also supported our previous structural hypothesis depicted from thermal data. For Raman measurements, additional glass compositions were prepared in the ternary system TeO_2_-Nb_2_O_5_-PbF_2_. Sample 3Nb0Pb of composition 97TeO_2_-3Nb_2_O_5_ presents the lowest niobium oxide content required for glass formation and represents a glass composition close to pure TeO_2_ which cannot be vitrified alone under classical quenching conditions. The other intermediary compositions 10Nb5Pb, 10Nb15Pb, and 10Nb25Pb were prepared for a better visualization and understanding of the Raman spectral changes. The spectrum of 3Nb0Pb is dominated by three broad signals centered at 450 cm^−1^, 660 cm^−1^ and 740 cm^−1^. These tellurite bands are well-known and largely described in the literature as being due to bending modes of Te-O-Te bridges, stretching modes of seesaw TeO_4_ units and stretching modes of pyramidal TeO_3_ units respectively [57,58,59]. In other words, the glass network of a “pure” TeO_2_ glass is built from 3D chains of interconnected TeO_4_ and TeO_3_. By adding 10 mol% of Nb_2_O_5_ (sample 10Nb0Pb), the Raman spectrum is very similar but with a shift of the band at 660 cm^−1^ to higher frequencies, mainly attributed to a richer niobium environment around TeO_4_ units. Under lead fluoride incorporation from 5 to 30 mol%, the ratio of the signal intensity at 740 cm^−1^ with respect to the other one at 660 cm^−1^ progressively increases, supporting the previously suggested modifier behavior of PbF_2_ with a related conversion of TeO_4_ to TeO_3_ units.

Oxyfluoride units TeOxFy are also probably formed since fluoride ions are known to decrease the glass connectivity through terminal fluoride bonds. Another Raman band centered around 870 cm^−1^ also appears for higher lead fluoride contents and is often attributed to terminal Nb-O bonds in distorted NbO_6_ units. It is worthy to note that the Nb_2_O_5_ content remains constant along the 10NbyPb series, but this signal is very weak for sample 10Nb0Pb when compared to 10Nb30Pb. This is another proof of the modifier behavior of lead fluoride, which progressively breaks the 3D covalent network and promotes the formation of terminal Nb-O bonds. One can also suggest that Nb-F terminal bonds could be formed but their Raman signal is probably close to the Nb-O one within the broad Raman band around 870 cm^−1^.

The whole optical window of these heavy metal fluorotellurite glasses was also investigated in the UV-visible-NIR-MIR range and is presented between 350 nm and 8 μm in Figure 5. In the UV-visible range, the UV cut-off, supposed to be governed by the bandgap energy, is best visualized in the inset of Figure 5 and it appears that lead fluoride incorporation extends the optical window from 400 nm to about 370 nm. Such behavior of an increasing bandgap energy can be understood by the formation of molecular orbitals with a higher bonding character in the valence band and molecular orbitals with a higher antibonding character in the conduction band correlated with stronger chemical bonds formed in the glass network by fluorine incorporation (terminal Te-F and Nb-F bonds). However, the absorption of other impurities from the row materials (mainly transition metals) in this wavelength range can also not be neglected. In the middle infrared part of the optical window, the transparency is limited to about 6 μm by the multiphonon absorption of the tellurite network. The strong absorption band centered at 3.2 μm is due to hydroxyl groups OH, probably present through Te-OH and Nb-OH terminal bonds. The wide optical window of these fluorotellurite glasses is a key parameter for optical applications up to 3 μm reached by silicate and phosphate glasses.

The crystallization properties were further investigated by comparing the thermal behavior of undoped and Eu^3+^-doped glass samples in the series 10NbyPb as presented by the DSC curves in Figure 6. Although Eu^3+^ doping does not modify the thermal properties in compositions 10Nb0Pb and 10Nb10Pb, as better seen by the glass transition temperatures in Table 1, the low temperature crystallization event is highly influenced by Eu^3+^ in the glass network: whereas crystallization could not be detected for the undoped composition 10Nb20Pb in the temperature range 250–450 °C, the doped glass composition exhibits a sharp and intense crystallization event around 355 °C and another weaker and broader event above 400 °C, such behavior being very similar to the thermal events detected for undoped glass compositions with 30 and 35 mol% PbF_2_ (Figure 2).

In the case of sample 10Nb30Pb, Eu^3+^ doping promotes a shift of about 14 °C of the first crystallization event to lower temperature whereas the high temperature crystallization peak remains unchanged. Hence, one should emphasize the nucleating behavior of europium for precipitation of the low temperature crystalline phase, which was already attributed to a fluoride or oxyfluoride phase. Nucleating species are usually used to promote a preferential volume nucleation with homogeneously distributed crystallites in the glass bulk. Such crystallization mechanism is required for the production of transparent glass-ceramics. As demonstrated in Figure 7a, heat-treatments for 1 h at Tx were able to crystallize only the low temperature phase through a phase separation mechanism since the high temperature crystallization event was not affected by such treatment. In addition, DSC measurements performed on powder samples with controlled grain sizes (Figure 7b) pointed out that the position and intensity of the first crystallization event is not dependent of the surface area, which is a characteristic behavior of volume crystallization.

The glass-ceramic samples 10Nb30Pb-TT and 10Nb30Pb-Eu-TT were also characterized for a better overview of their crystalline/glassy relative composition and transparency. X-rays diffraction patterns of both glass-ceramics are presented in Figure 8 and the diffraction peaks could be attributed to tetragonal lead oxyfluoride Pb_2_OF_2_, in agreement with our previous assumption that the intense low temperature crystallization event should contain fluorine. These thermal and structural data support a phase separation mechanism between Pb_2_OF_2_ and the residual glassy phase with a preferential volume crystallization as well as the ability of Eu^3+^ to act as a nucleating agent for a better control of the nucleation and growth steps. Based on these results, it is assumed that the same crystalline phase is precipitated in both samples but the number of nuclei for this phase growth is much higher in the Eu^3+^-doped glass. Indeed, Figure 9 presents the UV-visible-NIR attenuation spectra of both undoped and Eu^3+^-doped precursor glasses and final glass-ceramics between 350 nm and 2500 nm as well as the samples visual aspect. Along this wavelength range, attenuation of incident light through the samples is promoted by both scattering and absorption mechanisms. Whereas the undoped glass-ceramic exhibits a low transparency and increasing attenuation for shorter wavelengths, the Eu^3+^-doped glass-ceramic prepared under the same heat-treatment conditions is highly transparent and similar to the precursor glass samples. This increasing loss in transparency for sample 10Nb30Pb-TT is attributed to light scattering of the lead oxyfluoride crystallites related to larger crystallite sizes. It is inferred that Eu^3+^ promotes the formation of more nucleation centers in the doped sample with an associated lower overall growth rate and smaller lead fluoride crystallites with very low scattering effects.

Finally, the emission spectra of the Eu^3+^-doped glasses and glass-ceramic 10Nb30Pb-Eu-TT under excitation at 394 nm were recorded as shown in Figure 10. Experimental decay rates from the excited state ^5^D_0_ together with the emission spectra also allowed to access transition probabilities, theoretical radiative lifetimes and quantum efficiencies through the Judd-Ofelt theory [60,61] as resumed in Table 2. Along the 10NbyPb glass series, all optical parameters suggest a progressive richer fluorine environment around europium ions. In fact, emission intensity ratios between ^5^D_0_ → ^7^F_2_ and ^5^D_0_ → ^7^F_1_ progressively decrease for higher lead fluoride contents. As described in detail in many works, the intensity ratio between these two transitions gives access to the overall symmetry around Eu^3+^ ions since (a) ^5^D_0_ → ^7^F_2_ is a forced electric dipole transition which is hypersensitive to the europium surrounding and is forbidden for centrosymmetric sites and (b) ^5^D_0_ → ^7^F_1_ is a magnetic dipole transition weakly affected by the local symmetry. Thus, our decreasing experimental intensity ratios with increasing lead fluoride contents is related to a more symmetric first coordination shell around Eu^3+^ ions. Experimental emission lifetimes and quantum efficiencies accordingly increase along the composition series because of a lower local phonon energy and reduced non-radiative processes.

Once again, such behavior is in good agreement with a richer fluorine environment in these glasses, making glass composition 10Nb30Pb promising as a heavy metal oxyfluoride glass host for luminescent applications in the near and middle infrared. On the other hand, optical parameters in the doped glass 10Nb30Pb-Eu and glass-ceramic 10Nb30Pb-Eu-TT are almost similar, pointing out that the crystallization process does not significantly affect the europium neighborhood and luminescence properties. Based on the similar sharp emission bands for both glasses and glass-ceramics, it is inferred that europium ions could stay in the residual glassy phase during crystallization instead of migrating in the crystalline lead oxyfluoride phase. Taking once again in mind that the Eu^3+^ optical features are not affected by ceramization, one should propose that part of fluorine also remains in the glassy phase and surrounds the luminescent ions.

## 4. Conclusions

Glass formation was investigated in the ternary system TeO_2_-Nb_2_O_5_-PbF_2_ by melt-quenching and it was found that an optimum niobium oxide content of 10 mol% allowed lead fluoride incorporation up to 35 mol%. Synthesis conditions were investigated and optimized to minimize fluorine loss released as HF(g). The glass series with increasing PbF_2_ contents exhibits lower glass transition temperatures as well as a phase separation with precipitation of lead oxyfluoride Pb_2_OF_2_ for the most PbF_2_-concentrated compositions. Lead fluoride also acts as a glass network modifier through terminal fluorine bonds and associated conversion of seesaw TeO_4_ units to trigonal pyramids TeO_3_. The optical window of these fluorotellurite glasses ranges from 370 nm to 6 μm. In the most PbF_2_-concentrated glasses (20 mol% and 30 mol%), Eu^3+^ ions play the role of nucleating agents of the lead oxyfluoride phase, the crystallization mechanism of which is governed by volume nucleation. Hence, the nucleation and growth kinetics are better controlled in these Eu^3+^-doped samples and allowed the production of transparent glass-ceramics whereas the corresponding undoped glass-ceramics are almost translucent. Fluorine incorporation also enhances the luminescent properties of Eu^3+^ ions explained by a higher overall symmetry around the europium ions and associated higher excited state lifetimes and higher quantum efficiencies. Based on these considerations, it seems that these new lead fluorotellurite glass compositions are promising materials to be used as luminescent hosts as well as photothermorefractive materials with a wide infrared optical window.

## Figures and Tables

**Figure 1 materials-14-00317-f001:**
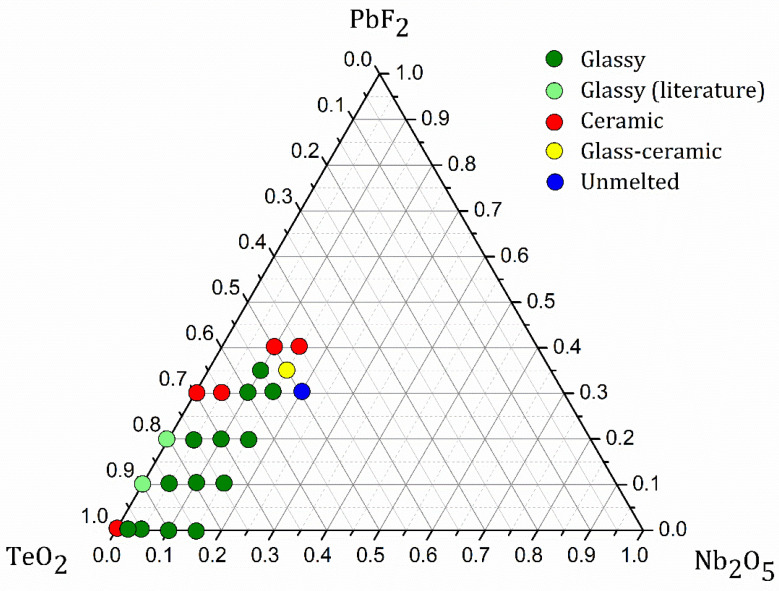
Glass-forming domain in the ternary system TeO_2_-Nb_2_O_5_-PbF_2_.

**Figure 2 materials-14-00317-f002:**
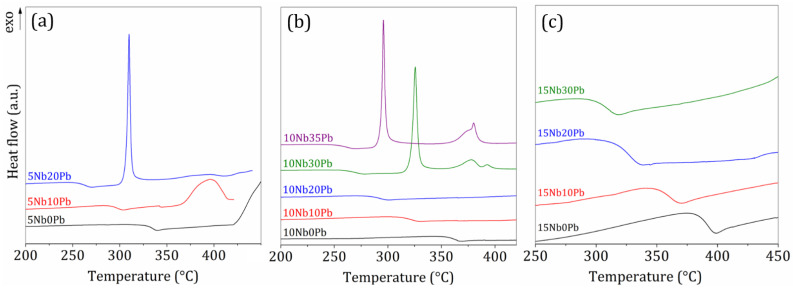
DSC curves of glasses in the glass series (**a**) 5NbyPb, (**b**) 10NbyPb and (**c**) 15NbyPb.

**Figure 3 materials-14-00317-f003:**
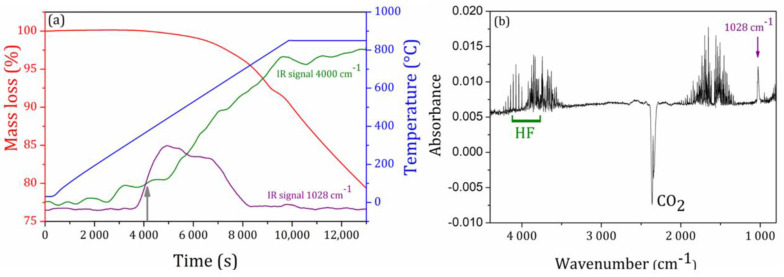
(**a**) Thermogravimetry of a starting powder mixture of composition 65TeO_2_-10Nb_2_O_5_-25PbF_2_, correlated absorption intensities of FTIR signals centered at 1028 cm^−1^ and 4000 cm^−1^ and (**b**) corresponding FTIR spectrum after 4130 s of measurement (grey arrow).

**Figure 4 materials-14-00317-f004:**
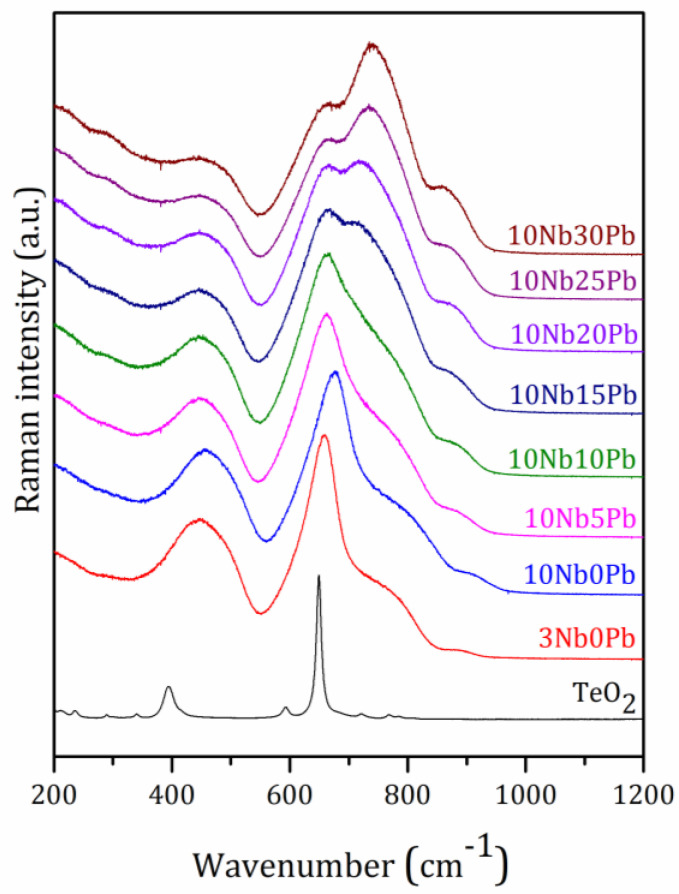
Raman spectra of glass samples in the ternary system (90 − x)TeO_2_-10Nb_2_O_5_-xPbF_2_.

**Figure 5 materials-14-00317-f005:**
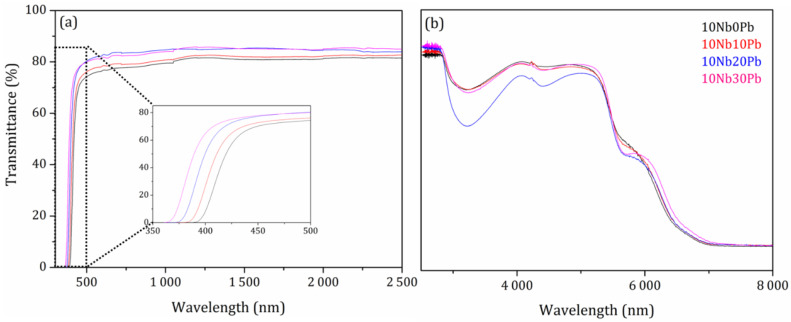
Optical window of glass samples in the (**a**) UV-visible-near infrared and (**b**) middle infrared ranges between 350 and 8000 nm.

**Figure 6 materials-14-00317-f006:**
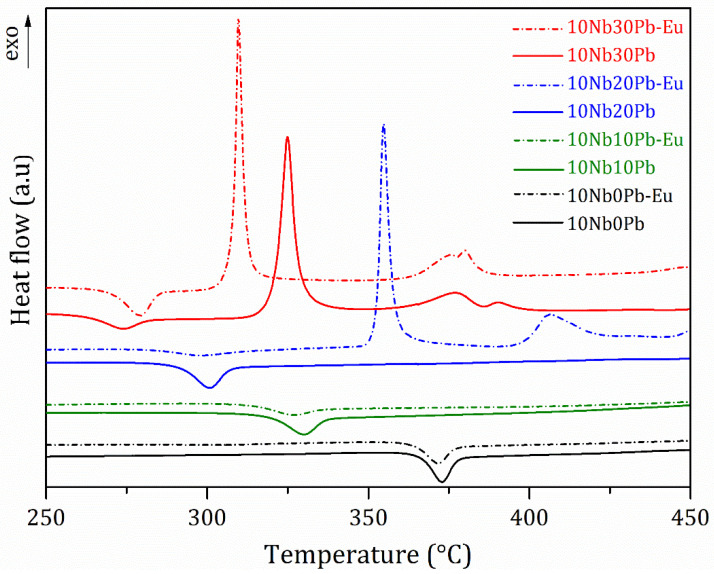
DSC curves of undoped and Eu^3+^-doped glass samples.

**Figure 7 materials-14-00317-f007:**
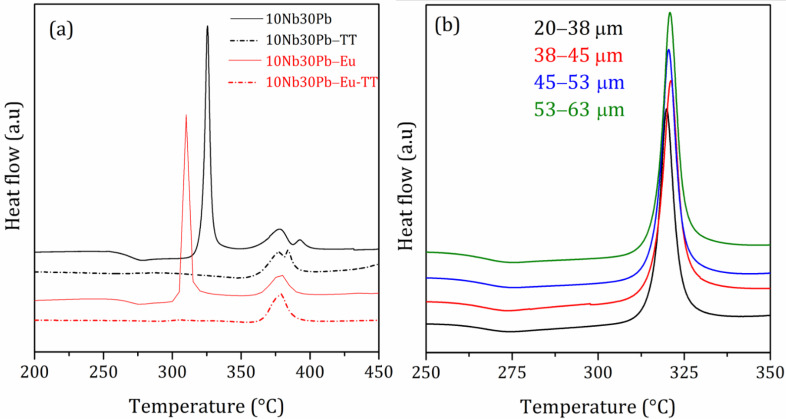
(**a**) DSC curves of samples 10Nb30Pb and 10Nb30Pb-Eu before and after heat-treatment and (**b**) DSC curves of sample 10Nb30Pb for different granulometries.

**Figure 8 materials-14-00317-f008:**
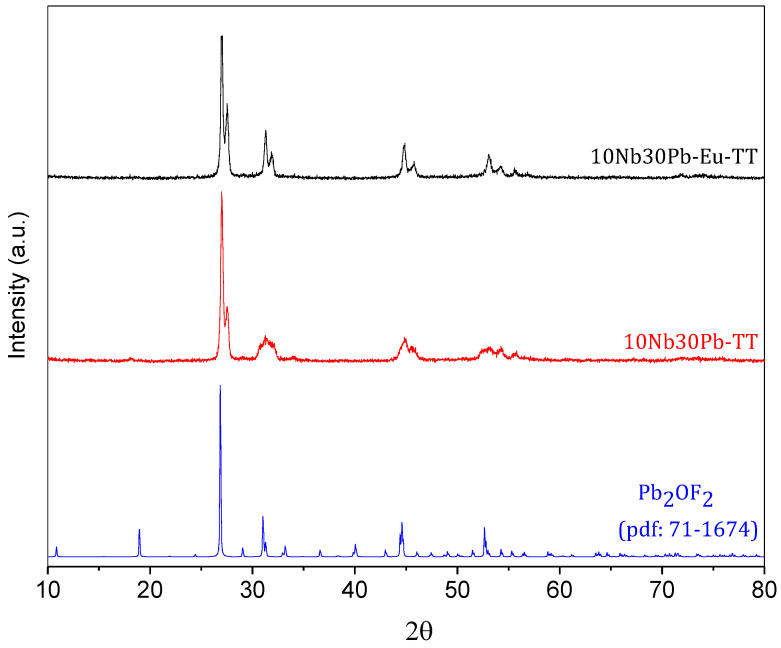
X-ray diffraction patterns of samples 10Nb30Pb-TT, 10Nb30Pb-Eu-TT, and crystalline reference Pb_2_OF_2_.

**Figure 9 materials-14-00317-f009:**
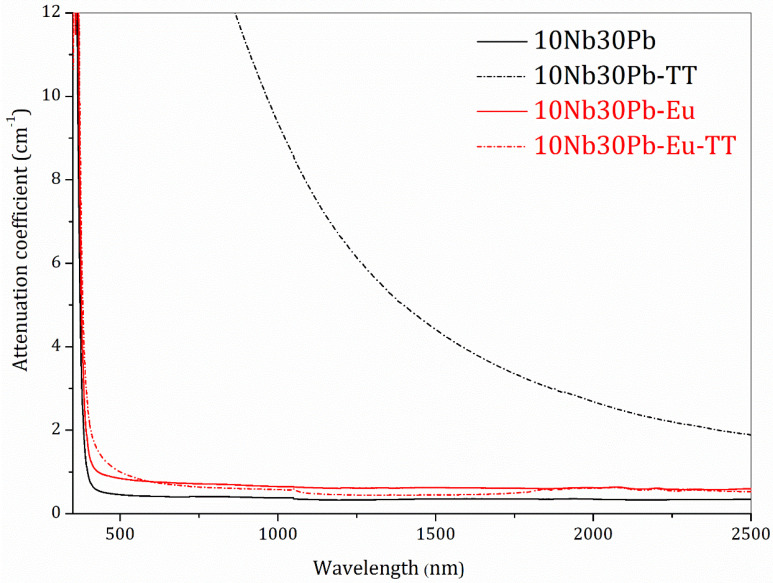
UV-visible spectra of glasses 10Nb30Pb, 10Nb30Pb-Eu, and glass-ceramics 10Nb30Pb-TT, 10Nb30Pb-Eu-TT.

**Figure 10 materials-14-00317-f010:**
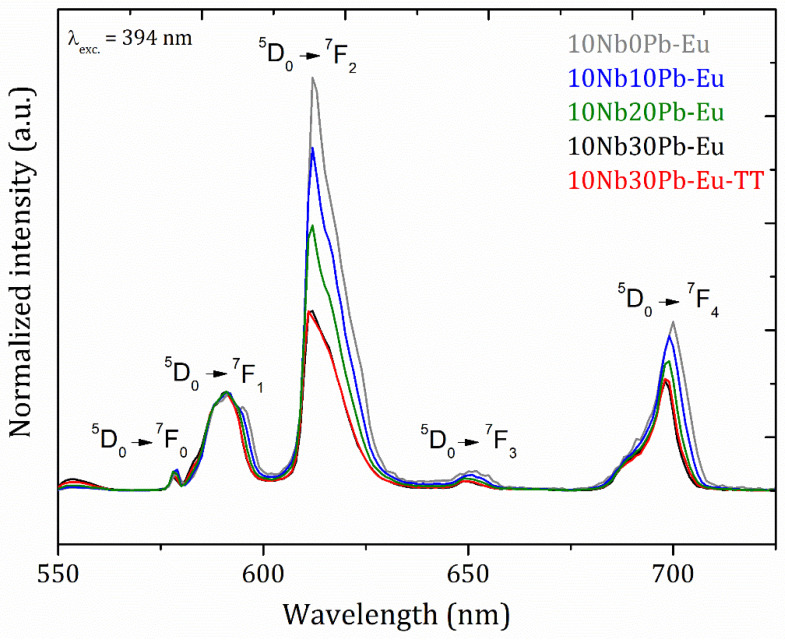
Emission spectra of Eu^3+^-doped glasses and glass-ceramics in the ternary system (90 − x)TeO_2_-10Nb_2_O_5_-xPbF_2_ under excitation at 394 nm.

**Table 1 materials-14-00317-t001:** Samples label, molar composition and characteristic temperatures.

Scheme	Composition (mol%)				Characteristic Temperatures (°C)			
	TeO_2_	Nb_2_O_5_	PbF_2_	Eu_2_O_3_	Tg	Tx_1_	Tc_1_	Tx_1_ − Tg
5Nb0Pb	95	5	0	-	335	424	453	89
5Nb10Pb	85	5	10	-	295	360	388	65
5Nb20Pb	75	5	20	-	265	307	310	42
10Nb0Pb	90	10	0	-	365	-	-	-
10Nb0Pb-Eu	89.9	10	0	0.1	365	-	-	-
10Nb10Pb	80	10	10	-	323	-	-	-
10Nb10Pb-Eu	79.9	10	19	0.1	324	-	-	-
10Nb20Pb	70	10	20	-	298	-	-	-
10Nb20Pb-Eu	69.9	10	20	0.1	296	354	357	58
10Nb30Pb	60	10	30	-	269	322	325	53
10Nb30Pb-Eu	59.9	10	30	0.1	271	308	310	37
10Nb35Pb	55	10	35	-	264	295	297	31
15Nb0Pb	85	15	0	-	390	-	-	-
15Nb10Pb	75	15	10	-	362	-	-	-
15Nb20Pb	65	15	20	-	327	-	-	-
15Nb30Pb	55	15	30	-	310	-	-	-

**Table 2 materials-14-00317-t002:** ^5^D_0_ → ^7^F_2_/^5^D_0_ → ^7^F_1_ intensity ratios, experimental lifetime (τ_exp_, ms) from ^5^D_0_ → ^7^F_2_ transition for Eu^3+^ ions, total radiative transition probability from level ^5^D_0_ (A_T_, s^−1^), radiative lifetime (τ**_rad_**, ms), and quantum efficiency (η, %) in Eu^3+^-doped glasses and glass-ceramic.

Sample	^5^D_0_ → ^7^F_2_/^5^D_0_ → ^7^F_1_ ^5^D_0_ → ^7^F_1_ (λ_exc_ = 464 nm)	τ_exp_ (ms)	A_T_	τ_rad_ (ms)	η (%)
10Nb0Pb-Eu	4.16	0.71	870	1.15	61
10Nb10Pb-Eu	3.46	0.92	767	1.30	70
10Nb20Pb-Eu	2.64	1.35	647	1.54	87
10Nb30Pb-Eu	2.00	1.75	535	1.87	93
10Nb30Pb-Eu-TT	1.98	1.67	551	1.81	92

## Data Availability

Data sharing is not applicable to this article.

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
