# Peer review of "Transparent Glasses and Glass-Ceramics in the Ternary System TeO2-Nb2O5-PbF2"

_materials, 2021, doi:10.3390/ma14020317_

Round 1
Reviewer 1 Report
Manuscript ID: Materials-1022108
Title: Transparent glasses and glass-ceramics in the ternary system TeO2-Nb2O5-PbF2
Authors: Juliana Santos Barbosa, Gislene Batista, Sylvain Danto, Evelyne Fargin, Thierry Cardinal, Gael Poirier *, Fabia Castro Cassanjes
Recommendation: Minor revisions
The manuscript reports synthesis, structural characterization, thermal and optical properties of ternary system TeO2-Nb2O5-PbF2 prepared by melt-quenching method. Presented results are very interesting, and the article is suitable for publication in this Journal. However, the minor revision is required for further consideration and some problems should be carefully corrected before publication:
- Manuscript needs some improvements in terms of discussing and comparing the obtained results with reported results in recent literature.
- In the manuscript is missing subscripts and superscripts in chemical formulas of compounds, please correct this.
- Eu3+ ions are usually employed as optical probes to investigate the surrounding environment. The authors analyze 5D0 → 7F2 / 5D0 → 7F1 intensity ratios (R/O ratios), but there is no information as to why this is so. R/O ratios was analyzed in detail in https://doi.org/10.1016/j.jallcom.2017.09.2173
- Descriptions in Figure 2 should be corrected in order to better readability of presented data (what’s this [u.a.]?).
- The reproducibility of preparing samples is missed in this manuscript, and the authors need to include this information in the revised version.
If above issues clarified in a good way, I would recommend the acceptance of this manuscript in the Materials.
Author Response
Reviewer´s comment: Manuscript needs some improvements in terms of discussing and comparing the obtained results with reported results in recent literature.
Author´s reply: the manuscript was revised and improved by discussing and comparing in a larger extent with recent literature.
Reviewer´s comment: In the manuscript is missing subscripts and superscripts in chemical formulas of compounds, please correct this.
Author´s reply: subscripts and superscripts were corrected.
Reviewer´s comment: Eu3+ ions are usually employed as optical probes to investigate the surrounding environment. The authors analyze 5D0 → 7F2 / 5D0 → 7F1 intensity ratios (R/O ratios), but there is no information as to why this is so. R/O ratios was analyzed in detail in https://doi.org/10.1016/j.jallcom.2017.09.2173
Author´s reply:
this sentence was added in the manuscript: As described in details in many works, the intensity ratio between these two transitions gives access to the overall symmetry around Eu3+ ions since (a) 5D0->7F2 is a forced electric dipole transition which is hypersensitive to the europium surrounding and is forbidden for centrosymmetric sites and (b) 5D0->7F1 is a magnetic dipole transition weakly affected by the local symmetry.
Reviewer´s comment: Descriptions in Figure 2 should be corrected in order to better readability of presented data (what’s this [u.a.]?).
Author´s reply: descriptions were corrected in Figure 2 for a better readability. [u.a] were corrected by [a.u] for arbitrary units.
Reviewer 2 Report
I find this work interesting, although some interpretations may have gone too far (line 193-211).
When it comes to material science aspects, my comments are as follows.
It is known that TeO2 glasses contain some OH groups because it is not so easy to get water-free TeO2 glasses.
In the work it was shown that HF ​​is present in the gas phase when glasses are heated. This gas is created by a reaction between PbF2 and H2O, with the formation of PbO. The decomposition of PbF2 to F2 and a reaction with Nb2O5 are less likely. In addition, one has to consider that not only TeO2 (approx. 450 Pa at 890 °C) but also PbF2 (approx. 2250 Pa at 890 °C) evaporate. For these reasons the glass contains less fluorine than was assumed. The TG curves show that when glasses are heated up to 850 ° C, a loss of mass is relatively large (> 10%) and is more than 20% when staying at this temperature. In summary, there are two fluorine compounds that volatilize in the system under investigation: HF and PbF2. Therefore a statement (line 193) that the nominal composition of glasses corresponds to the samples examined is incorrect. The X-ray measurement data from glasses was interpreted as Pb2OF2, although Pb2Te3O7 can also adapt very well.
The emission spectra of glass: Eu3+ are quite typical, although a short lifetime of around 1 ms makes it worth thinking about. In some work on the Eu-doped TeO2 glasses it was shown that both Eu3 + and Eu2 + can be present.

Author Response
Reviewer´s comment: In the work it was shown that HF ​​is present in the gas phase when glasses are heated. This gas is created by a reaction between PbF2 and H2O, with the formation of PbO. The decomposition of PbF2 to F2 and a reaction with Nb2O5 are less likely. In addition, one has to consider that not only TeO2 (approx. 450 Pa at 890 °C) but also PbF2 (approx. 2250 Pa at 890 °C) evaporate. For these reasons the glass contains less fluorine than was assumed. The TG curves show that when glasses are heated up to 850 ° C, a loss of mass is relatively large (> 10%) and is more than 20% when staying at this temperature. In summary, there are two fluorine compounds that volatilize in the system under investigation: HF and PbF2. Therefore a statement (line 193) that the nominal composition of glasses corresponds to the samples examined is incorrect.
Author´s reply: There are no evidence of PbF2 evaporation in a significative extent. In fact, the powder mixture is melted and the lead fluoride compound does not exist any more in the melt. We rather think that fluoride ions react with water or hydroxyls in the liquid state to form HF but without a significative lead loss. In fact, the final composition is different from the nominal composition with an expected lower fluorine content. For this reason, we assumed that the final glass composition is close to the nominal composition but we didn´t assume it is the same one. As explained in the manuscript (line 193), according to TGA analyzes the mass loss for a 5 min melting at 850ºC is estimated at 1,2%. For example, considering the starting powder composition 65TeO2-10Nb2O5-25PbF2 and if we assume that all fluorine is lost as HF, the final glass composition is 65TeO2-10Nb2O5-19PbF2-6PbO for a 1,2% mass loss, corresponding to a fluorine loss of about 23%. However, as suggested by the reviewer, TeO2 loss is also expected, giving a final glass composition closer to the nominal composition. The following sentence was added: Thus, for a starting powder mixture of composition 65TeO2-10Nb2O5-25PbF2 and considering that all fluorine is lost as HF produced by reaction between H2O or hydroxyls and PbF2, a 1,2% mass loss of the starting mixture gives the final molar composition 65TeO2-10Nb2O5-20PbF2-5PbO.
Reviewer´s comment: The X-ray measurement data from glasses was interpreted as Pb2OF2, although Pb2Te3O7 can also adapt very well.
Author´s reply: The reviewer is right and this attribution was also analyzed but it seems that diffraction peaks of Pb2OF2 match better our experimental diffraction peaks than Pb2Te3O7.
Reviewer´s comment: The emission spectra of glass: Eu3+ are quite typical, although a short lifetime of around 1 ms makes it worth thinking about. In some work on the Eu-doped TeO2 glasses it was shown that both Eu3 + and Eu2 + can be present.
Author´s reply: This hypothesis was checked but nor excitation spectra nor emission spectra exhibited broad bands which are usually characteristic of d-f electronic transitions of Eu2+ ions.
Reviewer 3 Report
I like this research. New glass compositions appear to be rather promising for photonics applications. The set of data obtained by the authors comprises the powerful starting point for future development of glasses and glass-ceramics based on this glass-forming system. I have no comments related to the experiments and their interpretation.
However, this manuscript cannot be published as it is.
My general impression is that the article was written very casually.
1) It seems to me that English should be improved.
Not to be unfounded, I will give several examples:
Through all manuscript text the authors write “… is related with…” and “attributed with …”. It would be more correct to write “… is related to…” and “attributed to …”.
Line 97. We read “…which allows for species detection in the range of… “. Allowing for TO ALLOW FOR being the same that, for example, TO TAKE INTO CONSIDIRATION, the expression used by the authors is not understandable. From the context it follows that the authors mean “…which allows detecting the species in the range… “. In the same line we read “decomposition glasses” – may be “decomposed glasses” is better?
Line 99. We read “…cold heat sinks…” What does it mean?
Line 110. We read “…using polishing sandpaper with decreasing grain sizes for a 1mm-thick final sample.” What does it mean?
Line 139 and others. We read “allow to incorporate…”. After ALLOW cannot be used an infinitive with “to” – only gerund. “Allow incorporating…” is correct.
Line 157. We read “…for the most PbF2 concentrated glasses …”. It seems to me that it means “the glasses with the highest concentration of PbF2”, does not it?
This list can be continued without my comments: “which are precisely released between 350ºC and 550ºC”, “specific reported molecular vibrations”, “F2 is produced from decomposition of PbF2”, “species could be at the origin of our IR signal at 1028 cm-1”, and other, other ,other examples.
2) The manuscript style is too heavy. It is advisable to divide the sections into subsections with subtitles. However, this is optional and at the discretion of the authors.
3) Several additional comments:
- the authors do not use superscripts and subscripts in chemical formulas and the designation of the wavenumber (cm-1).
- different transmission spectra are presented in different coordinates. It would be necessary to choose - either nm or cm-1.
- Line 125, the authors must specify the unit of temperature measurements.
Reviewer
Author Response
Reviewer´s comment: It seems to me that English should be improved.
Author´s reply: English was revised again by author and co-authors and improved according to the reviewer´s suggestions.
Reviewer´s comment: In the same line we read “decomposition glasses” – may be “decomposed glasses” is better?
Author´s reply: decomposition gasses replaced by "Evolved gaseous species"
Reviewer´s comment: what means cold heat sinks
Author´s reply: cold heat sinks replaced by cold surfaces
Reviewer´s comment: We read “…using polishing sandpaper with decreasing grain sizes for a 1mm-thick final sample.” What does it mean?
Author´s reply: replaced by “using polishing sandpaper until a 1mm-thick glass sample is obtained”
Reviewer´s comment: We read “…for the most PbF2 concentrated glasses …”. It seems to me that it means “the glasses with the highest concentration of PbF2”, does not it?
Author´s reply: “for the most PbF2 concentrated glasses” replaced by “for high PbF2 contents”
Reviewer´s comment: different transmission spectra are presented in different coordinates. It would be necessary to choose - either nm or cm-1.
Author´s reply: IR and Raman spectra were presented using wavenumber in cm-1 since this unit is usually used to express specific energies of vibrational modes. On the other hand, transmission and absorption spectra used to exhibit glass transparency in the UV-visible and NIR ranges were expressed in wavelengths (nm) for a more practical identification of the transparency window.
Reviewer 4 Report
- There are too many papers on the transparent fluorotellurite glasses, and regarding the research on the preparation and properties of this glass, it has no novelty. The main content of the manuscript is fluorotellurite glasses, so most of the contents can be deleted.
- There are also many reports about transparent tellurite glass ceramics. The luminescence intensity of Eu3+ doped glass ceramics was lower than that of glasses in Fig. 10, what is it meaningful to prepare the glass ceramics?
- The second part is too long and needs brevity.
- The XRD of figure 8 needs to be further analyzed; what are the similarities and differences of XRD between both samples 10Nb30Pb-Eu-TT and10Nb30Pb-TT?
- Their grain sizes need to be calculated. It is necessary to supplement TEM experiment to observe crystal phase and grain size.
- Figure 7 is best converted to transmittance. The absorption of Eu3+ cannot be seen in the figure. In this case, the error of result obtained by Judd Ofelt theory will be very large.
- Compared with figure 10 and table 2, it is not difficult to see that the lower the fluorescence intensity, the higher the quantum efficiency; Why?
Author Response
Reviewer´s comment: There are too many papers on the transparent fluorotellurite glasses, and regarding the research on the preparation and properties of this glass, it has no novelty. The main content of the manuscript is fluorotellurite glasses, so most of the contents can be deleted. There are also many reports about transparent tellurite glass ceramics.
Author´s reply: We agree that there are a lot of papers on transparent fluorotellurite glasses but based on the literature, there are only a few reports on transparent fluorotellurite glass-ceramics and no reports on transparent fluorotellurite glasses containing lead fluoride or oxyfluoride crystallines phases. In this sense, it appears to us that these transparent glass-ceramics are interesting for optics since the heavy metal is mainly inserted in the crystalline phase. The results on the glass samples in the first part of the work were presented for a better understanding of the whole procedure for glass-ceramics design.
Reviewer´s comment: The luminescence intensity of Eu3+ doped glass ceramics was lower than that of glasses in Fig. 10, what is it meaningful to prepare the glass ceramics?
Author´s reply: The luminescence intensity of Eu3+ glass-ceramics is not lower than that of glasses as demonstrated by the quantum efficiency in Table 2. Figure 10 only shows that the emission intensity ratio between both transitions decreases but these emission spectra were normalized at the maximum of emission band at 591nm. For this reason, luminescence intensities can not be compared from these data. In addition, we did not mean that glass-ceramics are better RE hosts than the starting glass but that the composition 10Nb30Pb is the most promising as RE host. On the other hand, it has been shown that Eu ions are important as nucleating agents for a more efficient nucleation step and final transparency of the glass-ceramics.
Reviewer´s comment: The XRD of figure 8 needs to be further analyzed; what are the similarities and differences of XRD between both samples 10Nb30Pb-Eu-TT and10Nb30Pb-TT?
Author´s reply: These XRD results were presented in order to highlight that the same crystalline phase is obtained in both samples (Pb2OF2). The difference in the Eu-doped sample is the more efficient nucleation mechanism and consequently smaller crystallites and more transparent glass. This sentence was added: “Based on these results, it is assumed that the same crystalline phase is precipitated in both samples but the number of nuclei for this phase growth is much higher in the Eu3+-doped precursor glass.”
Reviewer´s comment: Their grain sizes need to be calculated. It is necessary to supplement TEM experiment to observe crystal phase and grain size.
Author´s reply: Crystallite size could not be determined from DRX using the Scherrer equation since no broadening of the diffraction peaks were detected when compared to crystalline references. We fully agree that TEM experiments would be of great importance for size determination as well as phase composition (EDS). However, such experiments could not be performed during this work for technical reasons and access restrictions. We agree that these data could strengthen the work but we also thought that our presented results give a work of scientific quality.
Reviewer´s comment: Figure 7 is best converted to transmittance. The absorption of Eu3+ cannot be seen in the figure. In this case, the error of result obtained by Judd Ofelt theory will be very large.
Author´s reply: We did not understand this suggestion since Figure 7 presents DSC curves. In fact, Eu absorption is not seen in the absorption spectrum because of low concentration and low absorption cross-section. However, one should note that Judd-Ofelt parameters were not extracted from absorption spectra (as usual for other RE) but from emission spectra. In fact, The emission spectra of Eu3+ can be used because the 5D0/7F1 transition is only allowed via magnetic dipole and the transition probability of the other transitions can be derived from the intensity of the 5D0/7F1 transition band, as better explained in:
L.D. Carlos, R.A. Ferreira, V.Z. Bermudez, S.J. Ribeiro, Lanthanide-containing light-emitting organic-inorganic hybrids: a bet on the future, Adv. Mater. 51 (2009) 509e534, https://doi.org/10.1002/adma.200801635.
Reviewer´s comment: Compared with figure 10 and table 2, it is not difficult to see that the lower the fluorescence intensity, the higher the quantum efficiency; Why?
Author´s reply: As explained above, Figure 10 does not show a decrease of the emission intensity for higher PbF2 contents because all emission spectra were normalized at the maximum of band centered at 590nm (5D0->7F1) because this transition is very weakly dependent of the Eu surrounding. These emission spectra only point out a decrease of the intensity ratio between transitions 5D0->7F2 and 5D0->7F1. As described in the discussion section, the increasing quantum efficiency for higher PbF2 contents has been attributed to a richer fluorine environment around Eu ions and a lower local phonon energy. Such environment minimizes non-radiative relaxation processes.
Round 2
Reviewer 4 Report
The author has revised the manuscript, but there are still some problems as follows:
- Transparent glass ceramics containing Pb have been widely studied, but their future applications are limited due to the toxicity of Pb element. It is more important to replace Pb with other elements. In addition, TeO2-Nb2O5-PbF2 glass has been reported for a long time, and this part in the manuscript should be greatly reduced.
- Fluorine is easy to volatilize at high temperature. If the composition analysis was not done, the glass composition in the manuscript is inaccurate and unreliable. The contents added in 193~195 lines in the revised manuscript have no experimental basis. As a result, some explanations based on the composition were also unreliable. Please analyze the glass composition by X-ray fluorescence, or ICP, EDX, etc.
- The results of XRD were not consistent with the diffraction card of the reference crystal Pb2OF2.
- The absolute value of the integral intensity of the fluorescence for each sample need to be provided in Fig. 10 and insert it into Fig. 10 by the way of illustration.
- What are the refractive index of the samples?
- The Judd-Ofelt theory calculation should be added by deleting some contents about glasses.
- There is a big error in which measure the fluorescence intensity at different wavelengths by using Fluorescence spectrometer of different companies. If fluorescence intensity data were used in Judd-Ofelt theory, the reliability of the results is very low.
Author Response
Dear reviewer,
please find below the author´s reply to your notes:
1 - We agree that hasardous heavy metals are being progressively replaced by other environmentally friendly metals in materials science. However, lead is already interesting in low contents for specific applications where other metals don´t reach the same properties. From a general point of view, tellurium is also toxic but tellurite glasses are intensively investigated for specific optical applications. It is belived that solubility of both tellurium and lead is very low in atmospheric conditions and toxicity problems are related with the synthesis conditions. The authors decided not to remove the results concerning glasses in the system TeO2-Nb2O5-PbF2 since other three reviewers agreed with the relevance of this discussion part.
2 - We did not claim that final compositions are the same than starting compositions and the hypothesis described in lines 190-198 is only an estimation of final composition if we consider that all the mass loss is related with HF evaporation. The experimental basis is related with the clear HF detection by FTIR. This hypothesis shows the worst case for fluorine loss with a content variation of 5 mol%. If other compounds are also lost during melting the final glass composition will be closer to the nominal one. Since fluorine losses are often a critical point in oxyfluoride glass synthesis, we intended to show that fluorine loss can not br greater than 5 mol% in our experimental conditions. This assumption is based on the TG-FTIR experimental results.
3 - Based on our crystallographic study, it appears to us that diffraction peaks of our glass-ceramics and Pb2OF2 are found at the same positions but broadening of the peaks can be related with crystallites size. This is the closest crystallographic attribution using the JCPDS database.
4 - We still emphasize that absolute luminescence intensities are not reliable data since they strongly depend of the experimental conditions such as sample thickness, sample orientation, optical quality of the surface, model of fluorescence spectrometer etc... For this reason, we used intensity ratios as a more accurate parameter which is widely used to access optical properties.
5 - The refractive index of the samples vary between 2.14 (10Nb0Pb) and 2.11 (10Nb30Pb).
6 - The Judd-Ofelt theory calculation was not added to avoid repetition of well-known theoretical descriptions but the specific theoretical background of JO calculations based on emission spectra for Eu3+ can be easily found in references 60-61.
7 - We used only one fluorescence spectrometer but we agree that fluorescence intensity data are not reliable. This is exactly why we explained (reply 4) that absolute fluorescence intensity can be used as experimental data for calculations. However, the fluorescence intensity ratio is independent of such experimental parameters and is better used for JO calculations.